# Incorrect Facemask-Wearing Detection Using Convolutional Neural Networks with Transfer Learning

**DOI:** 10.3390/healthcare9081050

**Published:** 2021-08-16

**Authors:** Jesús Tomás, Albert Rego, Sandra Viciano-Tudela, Jaime Lloret

**Affiliations:** Instituto de Investigacion Para la Gestion Integrada de Zonas Costeras, Universitat Politecnica de Valencia, C/Paranimf 1, Grao de Gandia, 46730 Valencia, Spain; alremae@teleco.upv.es (A.R.); sandraviciano8493@gmail.com (S.V.-T.); jlloret@dcom.upv.es (J.L.)

**Keywords:** facemask-wearing condition, transfer learning, convolutional neural network, deep learning, facial recognition, COVID-19

## Abstract

The COVID-19 pandemic has been a worldwide catastrophe. Its impact, not only economically, but also socially and in terms of human lives, was unexpected. Each of the many mechanisms to fight the contagiousness of the illness has been proven to be extremely important. One of the most important mechanisms is the use of facemasks. However, the wearing the facemasks incorrectly makes this prevention method useless. Artificial Intelligence (AI) and especially facial recognition techniques can be used to detect misuses and reduce virus transmission, especially indoors. In this paper, we present an intelligent method to automatically detect when facemasks are being worn incorrectly in real-time scenarios. Our proposal uses Convolutional Neural Networks (CNN) with transfer learning to detect not only if a mask is used or not, but also other errors that are usually not taken into account but that may contribute to the virus spreading. The main problem that we have detected is that there is currently no training set for this task. It is for this reason that we have requested the participation of citizens by taking different selfies through an app and placing the mask in different positions. Thus, we have been able to solve this problem. The results show that the accuracy achieved with transfer learning slightly improves the accuracy achieved with convolutional neural networks. Finally, we have also developed an Android-app demo that validates the proposal in real scenarios.

## 1. Introduction

### 1.1. Motivation

In December 2019, cases of pneumonia of unknown origin appeared in Wuhan, Hubei, China, the clinical symptoms of which were very similar to pneumonia of viral origin. In the first group of infected people, it was determined that it was a zoonotic infection, that is, a viral transmission from animals to humans [1]. Due to the taking of samples from the lower respiratory tract, employing the RT-PCR technique in real-time, genome sequencing was carried out. This fact allowed us to clarify the taxonomy of this virus. New Coronavirus 2019 (2019-nCoV) was the name that was assigned to it.

Different studies have shown the spread by aerosols and fomites of SARS-CoV-2 among humans. In addition, it has been demonstrated that the virus can be spread even before the appearance of symptoms, referring to asymptomatic patients who are carriers of the virus, which implies that they can spread it without showing any symptoms of the disease. This fact has caused the rapid evolution and expansion of the pandemic. Due to the rapid spread of the virus, the World Health Organization (WHO) declared COVID-19 a public health emergency of international concern [2]. According to the WHO, as of 2 May 2021, there have been 151,803,822 confirmed cases of COVID-19, including 3,186,538 deaths.

The first symptoms found in this new coronavirus were coughing, fevers, and respiratory distress. However, with the progress of studies, other identifying symptoms of this virus have been determined. According to an update in March 2020 by the Council of State and Territorial Epidemiologists (CSTE), the loss of smell and taste was added as one of the compatible and most characteristic symptoms of this virus [3]. Moreover, in several studies, it has been shown that the pandemic has affected us mentally, not only because of the consequences left by the virus once the patient has recovered, but also because of the restrictions to which we have been subjected to reduce the transmission of SARS-CoV-2.

Mental health is associated with both demographic and psychosocial factors that, during this pandemic, cause a higher predisposition in some people to suffer these types of problems. The psychiatric disorders associated with this pandemic are usually stress, emotional disorders, depression, anxiety, sleeping problems, panic attacks, among others [4].

In addition to the symptoms of COVID-19, its rapid transmission between humans has been the object of study. According to studies, the contagion rate is higher in closed spaces without ventilation. This fact is because viral particles are capable of traveling in microdroplets (<10 μm), also called aerosols. These aerosols are produced by the human being when we speak, sing, laugh, etc. In addition, their speed increases with the force of the flow, for example, when we run or shout. What happens is that the largest drops fall to the ground, however, its nucleus (where the virus particle is located) is suspended in the air and is capable of being inhaled by another person, producing a possible infection [5,6].

Due to the transmission of the virus by air, many countries have introduced the mask as a mandatory use for protection against possible infections. With the use of the mask, the morbidity of COVID-19 and its associated mortality has been reduced. In addition, medical care has been reduced, preventing health systems from collapsing [7]. In addition, the use of the mask together with social distancing has managed to flatten the epidemic curve. The masks have the following two main functions: on the one hand, to prevent the viral particles that travel in the aerosol from being transmitted between the general population when coughing, speaking, sneezing, etc. On the other hand, the material from which the mask is made allows the volatile particles to be filtered from the air. In addition, the use of the mask has been of great help in preventing asymptomatic patients from infecting the rest of the general population [8]. Recent studies show that surgical masks effectively reduce the emission of viral particles. Coronavirus was detected in 30 and 40% of samples collected in participants without face masks, but no virus in droplets or aerosols was detected in participants with face masks. This study was conducted on exhaled air samples of SARS-CoV and MERS-CoV from infected patients and the findings indicate that surgical masks effectively reduce the emission of viral particles [9].

The recommendations of the Centers for Disease Control and Prevention (CDC) [10,11] indicate that while wearing the surgical mask, on exhalation, the air from the nose and mouth leaves with a high velocity and is directed frontally. The particles are relatively thick, between 3 and 8 microns (1 micron = 0.001 mm), and impact directly on the inside of the mask. Even if air escapes through the edges, bacteria, or other particles, do not escape since, due to their thickness, they are not able to follow the flow lines of the air that leaves the edges as long as the adjustment is correct.

The proper use of masks requires strict adherence to general hygiene measures, among which adequate coverage of the mouth and nose, avoiding gaps between the face and the mask, stands out. A partial, incorrect, or asymmetric fit poses a high risk for the transmission of infection [12].

It is due to the great importance of the use of masks that it is necessary to control their correct use. This fact leads to an increase in the control methods of non-pharmaceutical products that allow reducing the transmission of the virus [13]. For this reason, methods based on Artificial Intelligence (AI) have taken on great relevance, which allows a more exhaustive control over mask use in public spaces or areas with large population concurrence.

### 1.2. The Aim of the Study

Our objective is to study the application of Machine Learning techniques to distinguish whether a person is wearing a mask properly. Therefore, this project must take into account not only the presence or absence of a mask, but also its proper use, meaning, and being able to distinguish when it is well placed and when it is not. It offers detection and warning regarding multiple possibilities of placement errors that, without this application, would be hard for a person to notice. Although one of the most obvious mistakes is the use of the mask under the nose, there are others, such as the use of the mask over glasses, which are more difficult to detect, but no less important, or the poor fitting of the mask to the face. For the development of this application, deep learning techniques have been used to recognize people’s faces and proper mask use. For this aim, we need a solid training set that allows us to achieve good accuracy and reduce bias. Since there is no training set available for this task, one of the challenges of the project has been its creation.

### 1.3. Related Work

Due to the serious threat of the COVID-19 pandemic, novel solutions to optimize the incorrect use of prevention mechanisms are a hot topic. AI is one of the most useful techniques in adapting problems such as image classification to different situations. In this subsection, we are going to describe the most relevant works about AI and COVID-19 technical solutions.

AI and CNN solutions can solve problems by detecting patterns in images. Authors such as Chung et al., in [14], have developed an application integrated with the mobile phone capable of recognizing and classifying plants through the use of images using InceptionV3 CNN [15]. They have also implemented a prototype for identifying tree species with which real-time classification is performed remotely.

However, the current pandemic situation produced by SARS-CoV-2, in which we find ourselves, has led various authors to try to improve the control of the spread of the virus by developing applications to control the use of masks. Although, indeed, in previous years, authors such as Nieto-Rodriguez et al. had already developed this type of system to deal with other epidemics. Nieto-Rodriguez et al. in 2015 [16] developed a real-time image processing system in VGA resolution reaching 10 fps. VGA resolution allows the object to be 5 m from the camera to distinguish faces or masks. The system was developed to control the use of masks by medical personnel within operating rooms. In this way, an alarm goes off when the health personnel do not carry it because its use is mandatory.

Nevertheless, it has been in the last two years that the number of publications related to these AI systems has skyrocketed due to the critical situation we are facing.

Chen et al., in [17], have developed a mobile application that allows us to determine the service life of a facemask, indicating what period it is in, in addition to telling us what its level of effectiveness is after a period of use. To do this, they use microphotographs by extracting four characteristics of gray, employing co-occurrence matrices (GLCM) from the microphotographs of the facial mask. Using KNN, three results are obtained. The precision of these is 82.87% (macro measurements). The precision of “normal use” and “not recommended” reaches 92.00 and 92.59%.

Nonetheless, the need to control the overcrowding of people who wear or do not wear masks in public spaces has increased in importance in recent years.

Nagrath et al., in [18], have developed a design that can differentiate between the use of a facial mask or not. To perform real-time mask detection, they have used the MobileNetV2 architecture [19] as the framework for the classifier, together with the SSDMNV2 approach, which uses the single-shot multi-box detector as a face detector. By these means, they propose to use deep learning TensorFlow, Keras, and OpenCV to detect face masks. The precision of this study is 92.64% and it has an F1 score of 0.93. Mata, in [20], created a CNN model to be able to differentiate which people use a mask and which do not. It is based on a deep learning technique using an image or a video stream.

In a study carried out by Jauhari et al., in [21], the aim was to detect facial patterns to be able to detect the presence of facial masks in images. For this, it was based on Single Board Computer (SBC) Raspberry Pi. A face detection system based on the Viola Jones method was used to obtain efficient, fast, and accurate results. This method allows the adjustment of the cascade classifier to determine the area of the face in the image. The precision of this study is 90.9%. Sen et al., in [22], through the sequence of images and video, have developed a system capable of differentiating between people who wear face masks from those who do not. They use a MobileNetV2 model [19] along with python’s PyTorch and OpenCV for mask detection. The model has an accuracy of 79.24%. At the same time, an entry system to public places, which differentiates people who wear a mask from those who do not, has been proposed by Balaji et al. in [23]. In addition, this system has an alarm that emits a beep with a red or green tone to alert if a person is not wearing a mask. A Raspberry-PI camera is used to capture the video and transform it into images for further processing.

Recent studies show the applicability of these types of applications. Kurlekar et al., in [24], have developed a system that can be integrated with offices, airports, and public places in general. With their application, they can detect face masks in static images as well as in real-time videos. To do this, they used Computer Vision concepts and Deep Learning, using OpenCV and Keras/TensorFlow. Sakshi et al., in [25], using Keras/TensorFlow, developed a face mask detector. The architecture on which it was based is MobileNetV2 [19]. The model has been trained with several variations to ensure that the system can identify face masks in real time through video or still images. The final objective is, through Computer Vision, to implement the model in areas of high population density, health care areas, educational institutions, etc.

In 2020, Cheng et al., in [26], proved that the detection of the use of masks was important in stopping the spread of the virus. With the use of YOLO v3-tiny, it has proven to be suitable for the real-time detection of mask use. Plus, it is small, fast, and suitable for mobile hardware deployment, as well as real-time detection. Loey et al., in [27], developed a hybrid system for the detection of face masks. They selected three data sets. The simulated masked face data set, the real-world masked face set, and the tagged faces in nature. The design of this study is composed of Resnet-50 [28], for the feature extraction component. A second component for the classification of face masks is used by this system based on Support Vector Machines (SVM) and a joint algorithm for the mask classification process. The precision of the system is 99.49%, 99.64%, and 100%, respectively, for each of the data sets studied.

In addition to the need to be able to detect the use or not of a mask, the detection of when it is used in a wrong way due to its incorrect positioning is of great relevance. This misuse significantly reduces its effectiveness against the virus. For this reason, several authors, in addition to detecting its presence or absence, have focused on detecting its correct or incorrect placement. In 2020, Rudraraju et al., in [29], developed an application based on two steps. On the one hand, it detected the use or non-use of a facial mask. After detecting a mask, it distinguished between its correct or incorrect use. To do so, it relies on fog computing. Two nodes are used to process the video sequencing. Each fog node implements two MobileNet models [19]. For face detection, Haar cascade classifiers are used. Streaming takes place locally at each fog gateway without relying on the Internet. In this way, only the mask is allowed to enter the room and only if the mask is well placed. The accuracy of this system is around 90%.

Wang et al. 2021, in [30], using hybrid machine learning techniques, proposed to detect the use of masks using a two-stage approach. In the first stage, the user wearing a face mask is detected using the Faster RCNN and InceptionV2 [15] structure model. The second step is directed to a stage of verification of real face masks implemented by a classifier through a learning system. The general accuracy for simple scenarios is 97.32%, while for more complex scenarios it is 91.13%.

Smart Screening and Disinfection Walkthrough Gate (SSDWG) was created by Hussain et al. in 2021 [31]. It is a low-cost, fast and effective virus spread detection and control system based on IoT, for all places of entry. In addition to registering body temperature through temperature sensors that do not require physical contact, the system is also capable of differentiating people who wear face masks from those who do not. For the classification, it was also added not only if they were wearing a mask but also their correct use. For this classification, VGG-16, MobileNetV2 [19], InceptionV3 [15], ResNet-50 [28], and CNN have been compared using a transfer learning approach. The use or non-use of the mask was implemented through deep learning in real time. The obtained precision was 99.81 and 99.6% using VGG-16 and MobileNetV2, respectively. In addition, the classification of the type of mask, either N-95 or surgical masks, has also been implemented. Qin et al., in [32], using super-resolution and classification networks (SRCNet), with the training from the Medical Masks database, have developed a method to identify the presence or absence of a mask. This method, in addition, is capable of identifying the most frequent error of its misuse, such as wearing the mask under the nose. The algorithm used is based on the following four steps: image pre-processing, face detection and cropping, super-resolution images, and mask detection. The precision achieved with the use of this methodology is 98.70%. Table 1 summarizes the different works presented in this section.

The rest of the paper is structured as follows. In Section 2, materials and methods, we explain how we obtained the training data, how they are labeled, and the details of the intelligent system. In Section 3, the results are shown and described. Section 4 includes the discussion and, finally, Section 5, shows the conclusion along with future work.

## 2. Materials and Methods

This section describes the methods used in solving the problem. Our system is divided into 5 phases. Each one of the first five points in this section corresponds to these phases. In the last section, the system validation is described.

### 2.1. Obtaining the Data Set

The main objective of this paper is to identify mask placement errors using machine learning techniques. The drawback of machine learning is the need of a properly labeled training set.

Most of the works related to mask detection have chosen to use a synthetic corpus [33]. The idea consists in drawing, on the image of a face, the drawing of a mask. This method has been very useful in detecting if a mask is being used. However, the problem that interests us is much more complex. We want to detect small problems with the placement of the mask. The synthetic corpus would not work for our problem.

To tackle the problem, we decided to resort to citizen collaboration. We developed an application, shown in Figure 1, for mobile phones, which asked users to place the mask in different positions and take a selfie. The application was published on Google Play [34] and we went to the media to disseminate it [35]. The application was downloaded by more than 500 users during the summer of 2020 and about 3200 images were obtained, with a resolution of 360 × 480, half of them from the front and the other half from the side.

### 2.2. Labeling

The labeling was carried out by a nursing group from the hospital of Ontinyent. To speed up the work, an Android application was developed that allowed us to label 12 types of problems as well as the location where the problem was evident. Figure 2 shows this application, with an image from the front and another from the side. The labels were: 1—mask incorrectly extended, 2—non-symmetrical placement, 3—incorrectly bent in the nasal part, 4—adjusted below the bridge of the nose, 5—glasses placed under the mask, 6—neck adjustment greater than 1 cm, 7—with a beard, the use of a mask is not recommended, 8—incorrectly placed rubber band, 9—lateral gap greater than 1 cm, 10—exposed nose, 11—without a mask, 12—others.

Table 2 shows the statistics of the labeling process. Up to three errors could be indicated in each image. With the correct mask, 24.5% of the images were labeled, with an error of 55.9%; with two, 15.0%; and with three, 4.5%.

The application asked the user to take two selfies; one from the front and the other from the side of the face. As can be seen in Table 2, some errors were better detected from the side, such as “incorrectly placed rubber band” or “lateral gap greater than 1 cm”. However, it was decided that the side angle selfie would not be used given the user’s great difficulty in taking them. In fact, almost 10% of the side photographs were poorly framed and had to be discarded.

In this project, we will focus on the 5 most frequent problems (see Figure 3). To achieve this, some errors have been eliminated, such as “incorrectly placed rubber band” and “neck adjustment greater than 1 cm”. Others, such as “adjusted below the bridge of the nose” and “incorrect nasal bend” have been joined into a single type: “bad adjustment of nasal bridge”.

Only the images from the front and that also only present one type of problem, or none, will be used. These images have been divided into two sets, 1000 for training and 194 for validation.

It is important to highlight that there is high subjectivity in the labeling of the data set. Determining that the mask is perfectly fitted is relative. Some tests carried out show how two experts evaluating the same set provided a difference in up to 10% of the labels.

### 2.3. Facial Detection and Cropping

When obtaining the data set, the volunteers were asked to place their faces on a template. However, these indications were not followed very strictly. To normalize this situation, it has been decided to perform face detection, to eliminate the edges of the image without relevant information. Rapid Object Detection Using a Boosted Cascade of Simple Features [36] was adopted for facial detection, which has been shown to perform well in obtaining facial areas.

### 2.4. Classification

Recently, machine learning has experienced a breakthrough thanks to the emergence of Deep Learning. More specifically, CNN are the main ones responsible for this revolution. A convolutional network is structured hierarchically. The first layers are responsible for extracting generic features from the image such as edges or textures. The following layers use these previous characteristics to search for more specific characteristics. This process is continued for several layers until it is possible to detect characteristics with a high semantic value such as the detection of eyes or nose. Finally, a conventional neural network is used to perform the classification.

Although CNNs are widely used in natural language or audio processing tasks, several studies show that their use have obtained the best results in image recognition tasks. These results make the use of CNN in our problem, a natural choice. Nevertheless, in machine learning, obtaining the data set is the most complex part. The most common is having little data.

A widely used technique to obtain the most out of the data set is Data Augmentation [37]. It consists of making small modifications to the images such as small rotations, translations, and zooming in the input images to increase the variability of the training set. After several experiments, we verified that translation and zoom operations did not improve the results. We think it may be because the face is already cropped in the images. Finally, the training dataset was randomly rotated in a range of [−5°, 5°] and with a horizontal flip.

When few training samples are available, the Transfer Learning technique is quite useful. This method is based on using a model that was previously trained on a large data set, usually in a large-scale image classification task. This model will be used to customize this model for our task.

Transfer Learning is applied in two phases. First, we use the convolutional layers from the original model for feature extraction. The last layers, where the images are classified, are replaced to fix our problem. In the first phase the convolutional layers are fixed, only the classification layers are trained. In the second phase, known as fine tuning, all layers are unlocked, and the system is retrained. In this way, the extraction of characteristics fits our task.

We have, nowadays, a great variety of convolutional networks with dozens of layers already trained at our disposal. We can highlight MobileNet [19], Inception [15], ResNet [28], VGG [37], and Xception [38]. All these networks have been trained with the ImageNet corpus [39], a large data set with more than 14 million images where 20,000 different objects are recognized.

### 2.5. Decision System

This section proposes a decision system to detect, in real situations, errors in the use of masks. Depending on these errors, the system acts in the following different ways: by alerting errors, asking the user to solve the problem, etc.

The block diagram of the system is depicted in Figure 4. There are four different actors that constitute our proposal. The first one is the face detection module. This module presents a computer vision solution for face detection in real time. The second one regards classifying. The classifying module receives the input from the face detection module. Then, based on the system explained in the previous section, the classifying stage detects the errors in the facemask placement. The error and situation analysis module is included in third place. This module is an algorithm that evaluates the error given by the classifying system to select the most appropriate action. This is explained later in this section. The actuator module interacts with all the previous ones to create the warning or thanks the person for their correct use of the facemask.

Now, the only module that we need to specify is the decision system. Figure 5 shows the flow diagram of the algorithm. First, the warning level is initialized. Then, the module waits for an output, which is the probability of having a facemask placement error. When the module receives the output from the classifying module, it analyzes the predicted class with the highest probability, that is, the predicted error. Some error classes are more important than others, in which case they will be labeled as serious. For the most important errors, such as no mask detected, the system should ask the user to wear a mask, raise an alert, and block the entrance to the place if necessary. Some other errors are low-risk errors, which can be solved with a warning message to the user. When this happens, the face recognition module has to be started again. However, if there are continuous errors with the mask, due to the fact that the user does not want to wear it correctly, that would be treated as a serious error.

### 2.6. System Validation

To validate that the proposed method can be used in real situations, a real-time demonstration on a mobile application has been developed. The application has been developed on Android and can be downloaded from Google Play [40].

The device must be placed at the entrance of a public place, such as a hospital or educational center. Using the camera, the presence of a face will be detected, to isolate the area of interest as indicated in Section 2.3. Using a voice message, it will be indicated if a facemask-wearing problem is detected. Otherwise, the user will be thanked for its correct use.

Given the limited resources of a mobile device, it was decided to use a small and fast model. Specifically, MobileNet V2 [19] is used. As shown in the next section, very competitive results can be obtained using only 14 MB of device memory. As will also be depicted in the next section, some types of problems are not detected satisfactorily (specifically, “overlapping with glasses” and “bad lateral adjustment”). For this reason, these types of detection have not been included in the demonstration.

## 3. Results

To validate the proposal, we have carried out the experiments described in this section. Firstly, input images were down-sampled to 224 × 224. The Adam method was adopted as the optimizer. The network was trained for 20 epochs with an initial learning rate of 10–4 and with a learning rate dropping factor of 0.9. The batch size was 32. Transfer learning was applied for fine-tuning the same parameters, except the learning rate reduced to 10–5. The OpenCV and TensorFlow libraries were used. The link [41] shows the Python code used in each experiment.

### 3.1. Convolutional Neural Networks

We start by testing a traditional convolutional network. The results are shown in Figure 6. The two upper curves correspond to accuracy and the lower ones to loss. The following four convolutional layers have been used: 32 of 3 × 3, 44 of 5 × 5, 128 of 5 × 5, and 128 of 3 × 3. In each layer max-pooling (2 × 2) and ReLU activation function is used. For classification, 3 dense layers of 512, 256, and 6 neurons are used.

Figure 7 shows the Confusion Matrix obtained in the validation set. Each row of the matrix represents the instances in an actual class, while each column represents the instances in a predicted class. See Figure 3 for more detail on the categories. Therefore, the diagonal shows the samples that are correctly classified. For example, in this experiment, there are 25 samples labelled as “NO MASK”. There are 23 samples that have been correctly classified, one as “NOSE OUT”, and another as “CORRECT”.

In this case, we can confirm that there is a high accuracy except for in the “GLASSES” and “FIT SIDE” classes. “GLASSES” corresponds to “overlapping with glasses”, “N. BRIDGE” to “bad adjustment of the nasal bridge”, and “FIT SIDE” to “bad lateral adjustment”.

### 3.2. Transfer Learning

To illustrate the advantages of using Transfer Learning as the facemask-wearing condition identification network, we compare the more relevant models, including MobileNetV2 [19], Xception [38], InceptionV3 [15], ResNet-50 [28], NASNet, and VGG19 [37]. We compare features such as network size, the number of parameters, depth, and accuracy for both our task and the ImageNet task, as shown in Table 3. The first and the second rows represent the first experiment described in this section, with and without data augmentation. The rest of the rows are the different models of Transfer Learning with Data Augmentation. Depth refers to the topological depth of the network. This includes activation layers, batch normalization layers, etc. ImageNet accuracy refers to the accuracy obtained in the ImageNet task [42]. Figure 6 shows the accuracy of the transfer learning models. As can be seen in Figure 6 and Figure 8, the precision is very noisy, varying greatly from one epoch to the next. In order to better compare the results, the accuracy shown in Table 3 corresponds to the average of the last three epochs.

For each of the indicated models, their feature extraction layers have been used. After these, a layer of averagePooling2D is added and two dense layers of 512 and 6 neurons. The training is carried out in two phases. On epochs 1 to 20, the transfer model is locked and only the classification layers are trained. On epochs 21 to 40, fine-tuning is performed, unlocking learning of all the layers. Other configuration details are described in the Training details. Figure 8 shows the evolution of training for each model. The two upper curves correspond to accuracy and the lower ones to loss. The best results are obtained with VGG16. An improvement of 5% is observed for the results obtained with CNN.

If we analyze these results in more detail using the confusion matrix (Figure 9), we can observe how some kinds of errors such as “GLASSES” and “FIT SIDE” present lower accuracy than the others. However, the mislabeled samples have been reduced from the CNN experiment. Consequently, the accuracy is improved with this model.

Table 4 shows the classification results obtained, in the validation set, for each of the classes using the VGG16 model.

## 4. Discussion

Due to the great general concern in society about the pandemic caused by the new virus called SARS-CoV-2 and with it, the need to use masks, many authors have developed different applications to detect the presence or absence mask as well as their proper use. In Table 1, different projects are presented with the main objective of developing applications capable of detecting masks.

While our best results’ precision is, at first glance, lower than most of the related work, our system analyzes a more complex problem, and the lowered accuracy is due to this fact. By simplifying our results to “MASK” and “NO MASK”, our accuracy increases to 100%, as can be seen in Figure 9, as the 25 “NO MASK” samples are classified correctly, and all the other samples are classified as one of the “MASK” classes, even if the exact errors are not always detected.

By simplifying our results to “NOSE OUT”, “NO MASK”, and “CORRECT” we can also use the confusion matrix shows in Figure 9. In this case, the classes “GLASSES”, “FIT SIDE”, and “N.BRIDGE” are unified with “CORRECT”. If we obtain the new confusion matrix and perform the calculations, the new precision obtained is 97.4%. This result is comparable to those obtained in [29,30,31,32] and could be improved with specific training for these three classes.

On the one hand, references [18,19,20,21,22,23,24,25,26,27] developed a mask detection system that can differentiate between the presence or absence of a mask. On the other hand, references [29,30,31,32] are also able to detect if the nose is outside the mask. This circumstance would reveal an incorrect use of the mask, but as has been discussed in this project, it is only one of the possible incorrect uses.

In the case of our study, a system has been developed that is capable of detecting not only the presence or absence of the mask, but also different placement errors that current systems are not capable of detecting, such as the mask being placed below the nose, incorrect placement due to the use of glasses, that the nasal bridge of the mask is not correctly adjusted, or that the mask is too wide for the person, causing lateral gaps where the virus can easily enter.

To do this, different CNN-based deep learning techniques have been tested. However, the use of data augmentation does not appear to offer significant improvements, possibly due to the way the images are cropped. The transfer learning technique has been used to try to alleviate training shortages. We have tested the current most successful models. The results vary depending on the network used. The VGG16 model presents the best results (83.4% precision). This shows us that the knowledge of image processing can be used in the problem of detecting the correct use of the face mask. As library software, we have used OpenCV and TensorFlow. In addition, our system can detect with great precision other errors not usually considered in the placement of masks that have been mentioned above. By ignoring these errors, this misuse can help spread the virus.

Although in the labeling part of the corpus, more mask wearing problems were considered, in the present study we work with five types of errors. With the easiest of these to detect, such as the “no mask” or “nose out” errors, we have obtained a precision of 100 and 93%, respectively. The detection of an “incorrect adjustment of nasal bridge” error has a success rate of 75% and “incorrect lateral adjustment” a success rate of 73%. The type of error with the worst results is “overlapping with glasses” with 56%. This bad result may be due to a lack of examples in the training set.

Finally, in addition to studying the system, a mobile application has been developed. This application is accessible to all citizens and can be used to see the mistakes made regarding the placement of the mask in situ. In this case, as a classificatory model, we use MobileNetV2. This is because it demands fewer resources than others that were tested in this project and, thanks to this, it can be implemented in real time on current mobile phones, which is a requirement of our demonstration. Moreover, although the error detection precision is decreased, it is still high enough to be able to use the application to detect errors in the misplacing of a mask.

## 5. Conclusions

Identifying mask misuse is challenging. The limitation in the data sets is the main challenge. Data sets on mask wearing status are generally small and only identify the presence of masks. To solve the problem, we have carried out a campaign to collect images through an app, appealing to citizen participation. The samples obtained have been labeled by a group of health experts.

To our knowledge, no studies have been conducted on the identification of different misuses of masks through deep learning. The study carried out from [29,30,31,32] only detected the most obvious error, consisting of wearing the mask under the nose. Our proposal is capable of detecting, in addition to the previous mentioned issue, other types of problems, which occur very frequently. Even many of the users are unaware that they are using the mask incorrectly. However, we have not been able to detect other types of problems, such as (“incorrect lateral adjustment” and “glasses underneath”). For these cases, it is necessary to find an alternative approach or increase the number of training samples.

To validate that our proposal can be used in real situations, a real-time demonstration, an Android application, has been developed. It can be downloaded from Google Play [40]. The system is made to detect errors through the use of selfies. For this reason, for errors of bad lateral placement such as the crossing of rubbers, it would be good to teach the system to detect it with a lateral image. Although there really is a significant problem, it should be taken into account that a bad lateral adjustment indirectly causes an alteration in the frontal positioning of the mask, which can be detected with a frontal image. In this case, the system can be improved gathering more samples and teaching the system to detect, among other alterations, the crossing of the rubbers laterally.

The results support the possibility of its use in real circumstances, which makes it possible to prevent the spread of the pandemic. In future works, we want to enhance the study including other kinds of mask wearing problems and study the inclusion of other types of inputs to improve the accuracy of the “GLASSES” and “FIT SIDE” classes. On the other hand, and despite the fact that the application is capable of detecting different anomalies, it may be necessary to teach the system to differentiate between different components that currently make up the mask. This is because they have become another complement to our clothing. Many of the masks have sequins, other drawings ranging from simple squares to drawn smiles. An improvement in the system would be to verify that the application can detect these modified masks just as it does with surgical and FP2.

Furthermore, this system could be applied to different networks and scenarios. We could apply this to Smart Cities or Industrial Internet of Things environments to prevent security issues and decide when an alert should be raised.

## Figures and Tables

**Figure 1 healthcare-09-01050-f001:**
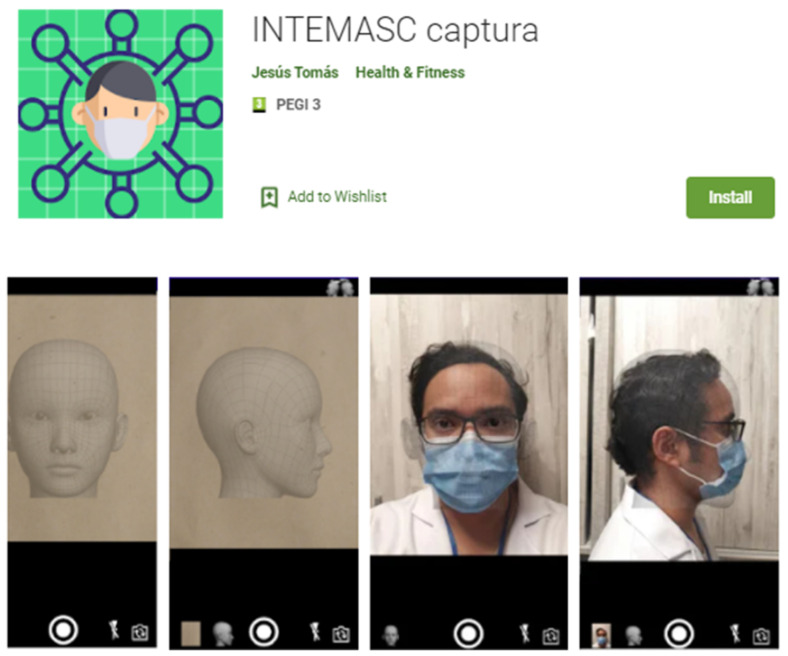
Application in Google Play for the acquisition of the training set.

**Figure 2 healthcare-09-01050-f002:**
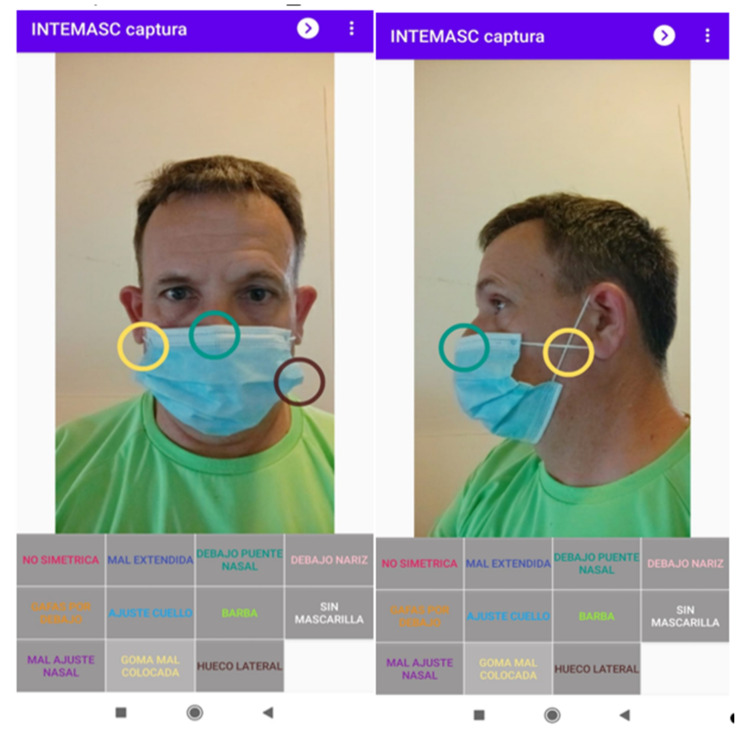
Application for the labeling of the training set.

**Figure 3 healthcare-09-01050-f003:**
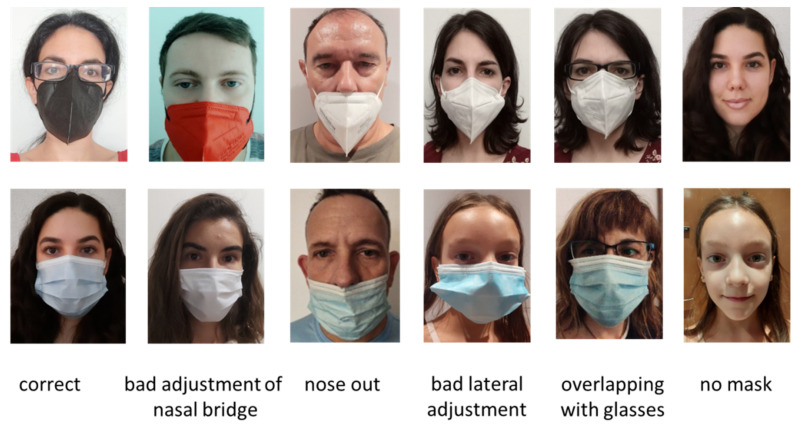
Example of samples of the 6 categories detected once cut.

**Figure 4 healthcare-09-01050-f004:**
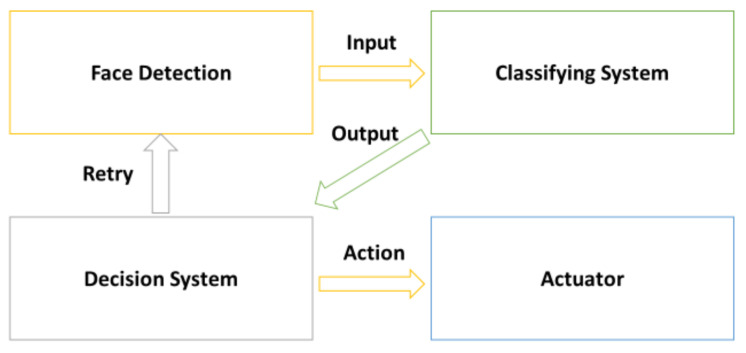
Block diagram of the system.

**Figure 5 healthcare-09-01050-f005:**
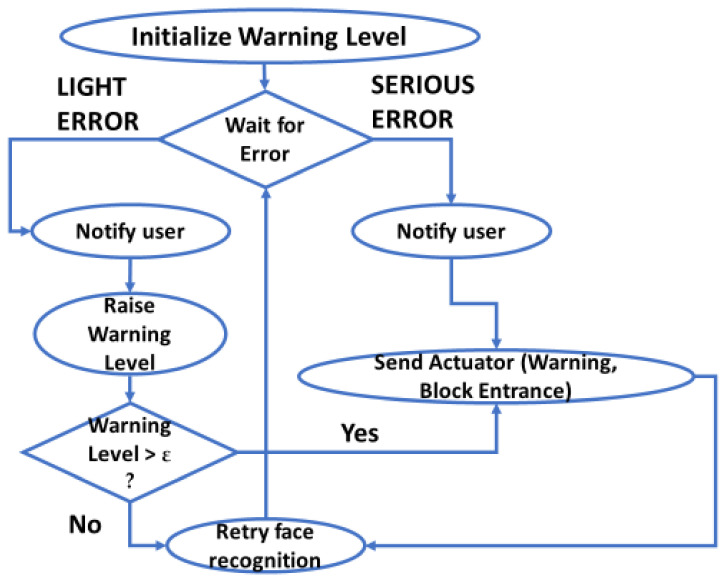
Flow diagram of the decision algorithm.

**Figure 6 healthcare-09-01050-f006:**
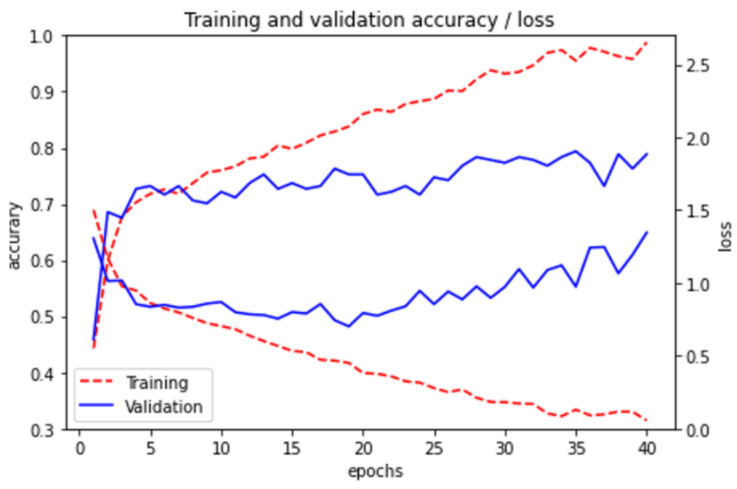
Accuracy and Loss for CNN model for training and validation set.

**Figure 7 healthcare-09-01050-f007:**
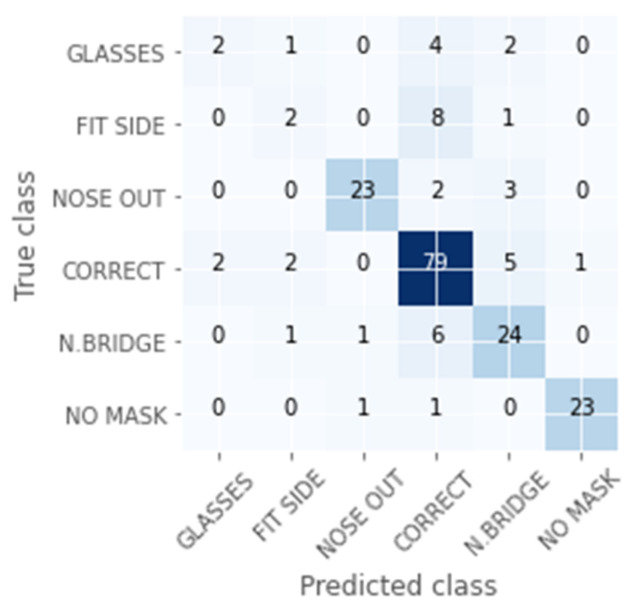
Confusion Matrix for CNN model in the validation set.

**Figure 8 healthcare-09-01050-f008:**
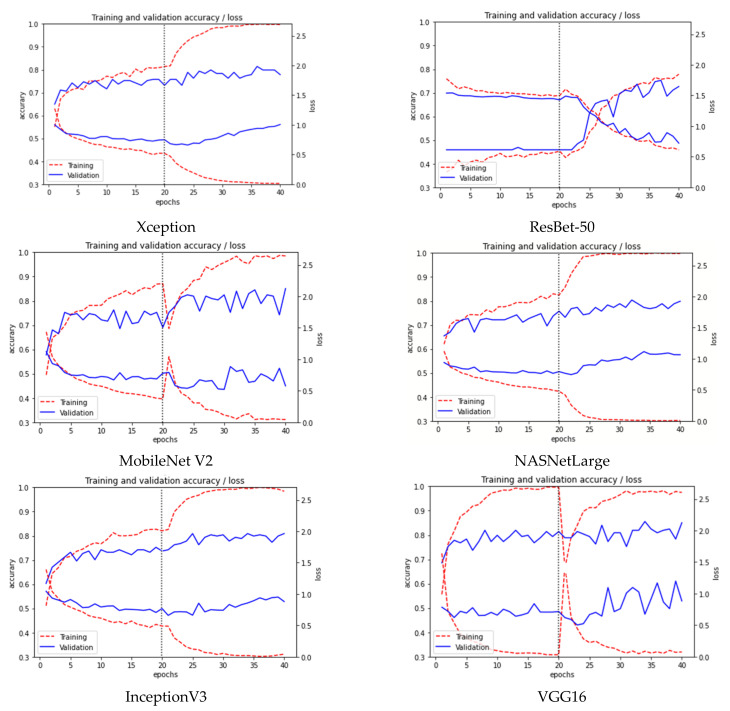
Accuracy for several Transfer Learning models for training and validation set.

**Figure 9 healthcare-09-01050-f009:**
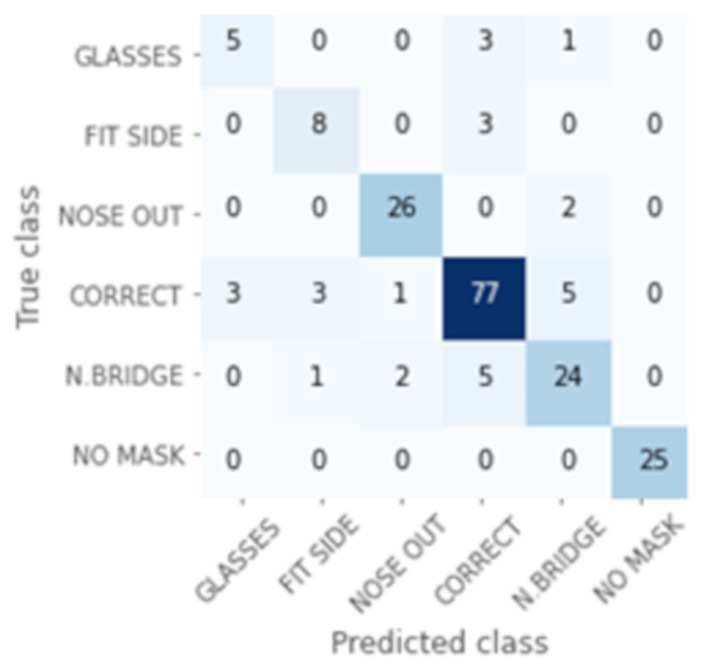
Confusion Matrix for VGG16 model in the validation set.

**Table 1 healthcare-09-01050-t001:** Summary of the related works.

1st Author [ref]	Date	Type of Detection	Face Detector	Classification Model	Software Library	Best Accuracy
Nagrath [18]	March 2021	mask/ no mask	Single shot multibox	MobileNetV2	TensorFlow, OpenCV	92.64%
Mata [20]	April 2021	mask/ no mask	Image Data Generator	CNN	TensorFlow, OpenCV	60%
Jauhari [21]	March 2021	mask/ no mask	Cascade Viola Jones	AdaBoost	Python	90.9%
Sen [22]	February 2021	mask/ no mask	-	MobileNetV2	PyTorch, OpenCV	79.2%
Balaji [23]	2021	mask/ no mask	Viola-Jones detector	VGG-16 CNN	TensorFlow, OpenCV	96%
Kurlekar [24]	April 2021	mask/ no mask	-	CNN	TensorFlow, OpenCV, Caffe	-
Sakshi [25]	March 2021	mask/ no mask	-	MobileNetV2	TensorFlow, Keras	99%
Cheng [26]	2020	mask/ no mask	YOLO v3 -tiny	CNN + SVM	-	-
Loey [27]	January 2021	mask/ no mask	YOLO v3	Resnet50 + SVM	-	99.5%
Rudraraju [29]	September 2020	mask/ no mask/ nose out	Haar cascade classifier	MobileNet	OpenCV, Keras	90%
Wang [30]	January 2021	mask/ no mask/ nose out	Fast RCNN	InceptionV2	OpenCV, Matlab	91.1%
Hussain [31]	April 2021	mask/ no mask/ nose out	YOLO v3	VGG-16, MobileNetV2, InceptionV3, ResNet50	Keras	99.8%
Qin [32]	September 2020	mask/ no mask/ nose out	Multitask Cascaded CNN	SRCNet	Matlab	98.7%

**Table 2 healthcare-09-01050-t002:** Statistics of the labeling process.

	Front Image			Side Image		
	1st Error	2nd Error	3rd Error	1st Error	2nd Error	3rd Error
Correct	518	0	0	476	0	0
without a mask	350	0	0	328	0	0
adjusted below the bridge of the nose	284	74	12	261	67	12
mask incorrectly extended	247	192	58	210	104	21
exposed nose	245	1	0	218	2	0
non-symmetrical placement	108	54	15	56	10	0
lateral gap greater than 1 cm	85	21	0	92	24	1
glasses placed under the mask	82	10	1	77	9	0
incorrectly placed rubber band	81	40	6	98	47	6
incorrectly bent in the nasal part	71	11	1	68	8	0
neck adjustment greater than 1 cm	25	9	2	34	12	2
with a beard, mask is not recommended	12	0	0	11	0	0

**Table 3 healthcare-09-01050-t003:** Comparison of the results obtained from the different models used.

Model	Size	Parameters	Depth	Accuracy	ImageNet Accuracy
CNN without data aug	32 MB	8.5 M	15	0.763	-
CNN	32 MB	8.5 M	15	0.797	-
MobileNet V2	14 MB	3.5 M	88	0.812	0.713
Xception	88 MB	22.9 M	126	0.802	0.790
InceptionV3	92 MB	23.9 M	159	0.819	0.779
ResNet-50	98 MB	25.6 M	-	0.742	0.749
NASNetLarge	343 MB	88,9 M	-	0.799	0.825
VGG16	528 MB	138.4 M	23	0.834	0.713

**Table 4 healthcare-09-01050-t004:** Classification results for VGG16 model in the validation set.

	Precision	Recall	F1-Score
GLASSES	0.56	0.63	0.59
FIT SIDE	0.73	0.67	0.70
NOSE OUT	0.93	0.90	0.91
CORRECT	0.87	0.88	0.87
N.BRIDGE	0.75	0.75	0.75
NO MASK	1.00	1.00	1.00
total	0.84	0.85	0.85

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
