# Peer review of "Incorrect Facemask-Wearing Detection Using Convolutional Neural Networks with Transfer Learning"

_healthcare, 2021, doi:10.3390/healthcare9081050_

Round 1
Reviewer 1 Report
The enclosed manuscript entitled "Incorrect Facemask-Wearing Detection Using Convolutional 2 Neural Networks with Transfer Learning" submitted by Tomas et al. implied an important computer-aided facial recognition procedure to justify if the masks are properly worn by the users. COVID-19 has been a global crisis that overshadows the entire world all over the economy, international travel, and even daily lives. To resume normal life, personal hygiene and mask-wearing become essential to safeguard the quick spread of the coronavirus. The authors intended to establish a CNN-based facial recognition system with a transfer-learning approach to overcome the lack of training sets in mask-wearing justifications. The rationale is an urgent need for social networks; however, there are some conceptual concerns prior to acceptance for publication.
1) The authors had made a comprehensive review in the background, including 13 studies of which 4 have put "nose out" in the assessment. Among the four reports, the accuracy range from 90% - 99.8% while the mask on-and-off discrimination remains very high accuracy. Despite the authors emphasized that the novelty of this study is to assess the misuse of masks in other aspects, but apparent lower accuracies were noticed in most assessments, rather than the "no mask" and "nose out." The authors may have to further elaborate on the novelty and make a more comprehensive comparison in the discussion.
2) The authors had suggested that the lateral image are relatively difficult to take by selfie, and it was mentioned to alternatively change the image acquisition by front view only. This could be a major pitfall in the justification of rubber band uses and side fitting. It is unknown how the authors would like to overcome the issue as these tests could be novelties against the previous studies.
3) The authors have spent a few contents discussing the use of other types, e.g. N-95, color, or other accessories, of masks that potentially affect computer-aided recognition. The authors may address this concern in the discussions.
4) There are different alternative use of masks, such as crossover of rubberbands and stereo folding of the mouthpiece, to create an air-tight zone for customized use. It is unknown if the assessors for the training sets made consensus justifications for the "correct use" of masks. The inter-rater correlation may be considered.
5) Despite the use of the VGG16 model provides better accuracy, it seems that the model is more calculating resource-demanding. It is unclear whether this concern is an issue affecting the application for public use.
6) A minor issue is that line 134-142 is duplicated.
Author Response
Thanks for your comments, we really appreciate them. These comments really help us to improve our work.
1) The difference in precision obtained between the proposed system and the previous works is due to two main factors: we detected more classes, and the training set is different. To clarify this concept, the following paragraph has been added after the first paragraph of the discussion section:.
While our best results’ precision is at first glance lower than most of the related work, our system analyzes a more complex problem, and the lowered accuracy is due to this fact. By simplifying our results to “MASK” and “NO MASK”, our accuracy in-creases to 100%, as can be seen in Figure 9, as the 25 “NO MASK” samples are classified correctly, and all other samples are classified as one of the “MASK” classes, even if exact errors are not always detected.
By simplifying our results to “NOSE OUT”, “NO MASK” and “CORRECT” we can also use the confusion matrix shows in Figure 9. In this case, the classes “GLASSES”, “FIT SIDE” and “N.BRIDGE” are unified with “CORRECT”. If we obtain the new confusion matrix and perform the calculations, the new precision obtained is 97.4%. This result is comparable to those obtained in [29-32] and could be improved with specific training for these three classes.
2) We take this annotation as an important piece of information to improve the system and to be able to add new forms of detection. The justification for why it has not been done is reflected in the line 534:
“In addition, the system is made to detect errors through the use of selfies, for this rea-son for errors of bad lateral placement such as the crossing of rubbers, it would be good to teach the system to detect it with a lateral image. Although there really is a significant problem, it should be taken into account that normally a bad lateral adjustment indirectly causes an alteration in the frontal positioning of the mask, which can be detected with a frontal image. For this, the system will be improved with a greater number of samples and teaching the system to detect, among other alterations, the crossing of the rubbers laterally. “
3) We really appreciate this comment, which help us to improve the quality of our paper. We have added this paragraph to solve this question in line 545:
“On the other hand, and despite the fact that the application is capable of detecting different anomalies, it may be necessary to teach the system to differentiate between different components that currently make up the mask. This is because they have become another complement to our clothing. Many of the masks have sequins, other drawings ranging from simple squares to drawn smiles. An improvement in the system would be to verify that the application can detect these modified masks just as it does with surgical and FP2 .”
4) Thanks for the comment. We have already taken this fact into consideration since it was an important problem during the development of our project. The data was evaluated by several health experts who determined which collocation problems we should address. Despite this fact, different evaluators could obtain different label sets. This is discussed in the following paragraph at the end of subsection 2.2 (line 289):
“It is important to highlight that there is high subjectivity in the labeling of the data set. Determining that the mask is perfectly fitted is relative. Some tests carried out show how two experts evaluating the same set provided a difference in up to 10% of the labels.”
5) Thanks for your help and your comments. This fact is indeed a key factor. In our demo, we use MobileNet to reduce the resources needed. This explanation has been added at the end of section 4.
“In this case, as a classificatory model, we use MobileNetV2. This is because it demands fewer resources than others that were tested in this project and, thanks to this, it can be implemented in real time on current mobile phones, which is a requirement of our demonstration. "
6) Thanks for notifying this issue. We have reviewed the entire document to be able to correct this error and any other errors of this type that might have occurred.
Reviewer 2 Report
This article presents an enhanced facemask detection system (followed by an application) using Transfer Learning. The work is excellent, well structured, balanced and I suggest to be accepted.
My minor comments:
Line 164-165. Real-Time Raspberry, I suggest Single Board Computer (SBC) Raspberry Pi
Line 5. please write your affiliation in English
Line 238. In Section 3, results ... please write a complete sentence.
Line 289. typo pla-ced
Line 326. typo zoom ... in the
Line 375. Improve Figure 5
Line 414. Starting a sentence with a number...
Line 466. typo
Line 507 figure 9 to Figure 9
Author Response
Thanks for your comments, we really appreciate them. These comments really help us to improve our work. We have fixed all these issues. Moreover, we have fixed some few more we have found in the text.